# Assessing Coagulation Parameters in Healthy Asian Elephants (*Elephas maximus*) from European and Thai Populations

**DOI:** 10.3390/ani12030361

**Published:** 2022-02-02

**Authors:** Sónia A. Jesus, Anke Schmidt, Jörns Fickel, Marcus G. Doherr, Khajohnpat Boonprasert, Chatchote Thitaram, Ladawan Sariya, Parntep Ratanakron, Thomas B. Hildebrandt

**Affiliations:** 1Department of Reproduction Management, Leibniz Institute for Zoo and Wildlife Research, 10315 Berlin, Germany; hildebrand@izw-berlin.de; 2Department of Evolutionary Genetics, Leibniz Institute for Zoo and Wildlife Research, 10315 Berlin, Germany; aschmidt@izw-berlin.de (A.S.); fickel@izw-berlin.de (J.F.); 3Institute of Biochemistry and Biology, University of Potsdam, 14476 Potsdam, Germany; 4Institute for Veterinary Epidemiology and Biostatistics, Freie Universität, 14163 Berlin, Germany; marcus.doherr@fu-berlin.de; 5Center of Elephant and Wildlife Health, Faculty of Veterinary Medicine, Chiang Mai University, Chiang Mai 50100, Thailand; khajohnpat@gmail.com (K.B.); chatchote.thitaram@cmu.ac.th (C.T.); 6The Monitoring and Surveillance Center for Zoonotic Diseases in Wildlife and Exotic Animals, Faculty of Veterinary Science, Mahidol University, Nakhon Pathom 73170, Thailand; ladawan.sar@mahidol.edu; 7Faculty of Veterinary Science and Applied Zoology, Chulabhorn Royal Academy, Bangkok 10210, Thailand; parntep.rat@cra.ac.th; 8Faculty of Veterinary Medicine, Freie Universität, 14163 Berlin, Germany

**Keywords:** coagulation, Asian elephant, EEHV, factor VII, F7 gene, prothrombin, activated PTT, fibrinogen

## Abstract

**Simple Summary:**

Asian elephants (*Elephas maximus*) are considered endangered and their population is in continuous decline. Understanding their social interactions, health, and welfare status has been a topic of intense research in recent decades. Coagulation assessments have been underutilized in wildlife but can give valuable information on individual health. This study aims to increase the knowledge of the coagulation status in healthy Asian elephants from different backgrounds and age groups, using a fast point-of-care analyzer. This tool can be further used in either routine health check-ups performed by caretakers or in a clinical emergency, such as in cases of elephant endotheliotropic herpesvirus hemorrhagic disease outbreaks. We have also investigated the presence of genomic mutations in one coagulation factor—factor VII—where a disorder was previously reported in an Asian elephant. Hereby, we report new reference values for coagulation parameters, such as coagulation times and fibrinogen concentration of Asian elephants assessed in Thailand and in Europe, as well as several single point mutations found in the exons of *Elephas maximus* coagulation F7 gene. We found the point-of-care analyzer used in this study to be very practical and user friendly for a zoo and field environment and hope that this project will incentivize further coagulation studies in Asian elephants and in other wildlife species.

**Abstract:**

The Asian elephant population is continuously declining due to several extrinsic reasons in their range countries, but also due to diseases in captive populations worldwide. One of these diseases, the elephant endotheliotropic herpesvirus (EEHV) hemorrhagic disease, is very impactful because it particularly affects Asian elephant calves. It is commonly fatal and presents as an acute and generalized hemorrhagic syndrome. Therefore, having reference values of coagulation parameters, and obtaining such values for diseased animals in a very short time, is of great importance. We analyzed prothrombin time (PT), activated partial thromboplastin time (aPTT), and fibrinogen concentrations using a portable and fast point-of-care analyzer (VetScan Pro) in 127 Asian elephants from Thai camps and European captive herds. We found significantly different PT and aPTT coagulation times between elephants from the two regions, as well as clear differences in fibrinogen concentration. Nevertheless, these alterations were not expected to have biological or clinical implications. We have also sequenced the coagulation factor VII gene of 141 animals to assess the presence of a previously reported hereditary coagulation disorder in Asian elephants and to investigate the presence of other mutations. We did not find the previously reported mutation in our study population. Instead, we discovered the presence of several new single nucleotide polymorphisms, two of them being considered as deleterious by effect prediction software.

## 1. Introduction

The world Asian elephant population faces several threats, especially in their range countries, including hunting, logging, loss of habitat, and consequent human–elephant conflict. According to the IUCN Red List, the number of Asian elephants has declined by ~50% over the last three generations [1]. Captive and wild elephant health status research is therefore paramount to aid the conservation efforts for this species. Elephant blood analytical examinations most often focus on blood biochemistry and hemogram tests. Assessment of coagulation parameters is virtually never used as part of a normal clinical check-up, and very rarely are they tested before surgery or other similar invasive interventions, due to the need of specific instruments and specialized operators.

In addition to a complete blood count, coagulation time results and fibrinogen concentrations can be used as valuable and easily accessible health indicators, because stress, illness, injury, medications, and surgery affect coagulation parameters [2]. Coagulation times provide information in a large variety of clinical ephemerons alterations, such as sepsis, hepatic disfunction, decrease in vitamin K, shock, trauma, embolism, platelet bleeding disorders, coagulation factory deficiency, and disseminated intravascular coagulation (DIC) [2,3,4,5]. Liver disfunction may affect the coagulation cascade in several ways since this organ produces most of the coagulation factors and affects vitamin K absorption. Therefore, any illness affecting the liver, such as inflammation, neoplasia, biliary statis, and the use of chronic medication, may lead to coagulation deficiency. Infectious diseases, severe systemic diseases, or immune-mediated diseases can also alter normal coagulation times. Due to this panoply of factors that may affect coagulation, it is suggested that coagulation times should be accessed as a pre-surgical test for any animal, regardless of age [2]. Fibrinogen is used as a specific and sensitive marker for inflammation in humans [6] and in horses, for example, and its early recognition has been shown to be essential for the diagnosis of diseases and proper treatment planning. In horses, fibrinogen serial testing provides information regarding treatment efficacy in length and prognosis in several infectious or inflammatory conditions, such as pleuropneumonia, abdominal abscess, endometritis, and endocarditis [2]. Coagulation is a process activated after a vessel damage, when the body reacts in order to stop the hemorrhage, locally creating a viscous and thick material—a clot—to seal this lesion [5,7,8]. Platelets start to adhere to the subendothelium, forming a plug, and sequentially activated intervening factors (coagulation factors) start interacting in a so-called “cascade”, in order to produce fibrin. Fibrin fibers form a mesh over the platelets, creating a seal at the injury site to stop further blood loss [5,7]. The regulation of this process is fine-tuned in order to control the growth of the clot and to prevent the aggregation of a thrombus, which can lead to complications such as stenosis or embolism [5,7]. Therefore, coagulation is a dynamic process between coagulation-promoting mechanisms and those that stop it from expanding beyond the injury site. Such maintenance of hemostasis is essential to avoid both continuous bleeding and thrombosis [3,7].

The “cascade model of coagulation” is the model most traditionally used to explain the complex process of clot formation. According to this model, a stepwise enzymatic conversion of zymogens (precursors that circulate in an inactive form in the plasma) leads to the final product, a fibrin clot. This synchronized enzymatic activation along the coagulation cascade splits into two main pathways: the extrinsic pathway (after vessel wall damage, it includes tissue factor and factor VII) and the intrinsic pathway (involving contact with a negatively charged surface and coagulation factors V, VIII, IX, XI, and XII). Both pathways then converge in the activation of factor X, leading to a final common pathway where fibrinogen is converted into fibrin [3,7,9]. The time needed for clot formation can be measured using the prothrombin time (PT) for the extrinsic pathway and using the activated partial thromboplastin time (aPTT) for the intrinsic pathway [5].

In humans, numerous genetic mutation(s) of the F7 gene, the gene encoding coagulation factor VII, are known to cause deficiency and reduced activity of this factor, leading to an overall reduction in the coagulation efficiency. The condition can be inherited or acquired, transmitted with autosomal recessive inheritance, and is among the rare congenital bleeding disorders—it is the most commonly present [10]. The most common type of mutation is point mutation (single-nucleotide polymorphism, SNP), which can either be silent (i.e., synonymous) when the coded amino acid (aa) sequence stays the same, or it can be a missense variant (i.e., non-synonymous) when the change causes an alteration in the aa sequence of the encoded protein. In humans, 221 unique variants have been reported so far for the F7 gene [11,12]. People with factor VII deficiency may experience prolonged and uncontrolled bleeding episodes with initial onset and bleeding severity varying greatly among people. While some individuals are asymptomatic, others may develop mild, moderate, or even severe life-threatening complications as early as in infancy [10]. As in humans, factor VII deficiency has also been reported several times in dog breeds, such as Beagles, English Bulldogs, Alaskan Malamutes, Boxers, and also in mixed-breeds [13]. Like humans, the symptoms of this deficiency also vary in dogs, as there the disease is normally not accompanied by spontaneous bleeding, although some animals present bruises and prolonged bleeding after surgical intervention [13]. In 2017, a factor VII deficiency was also detected in an Asian elephant bull, and, although the animal did not have a bleeding tendency, it demonstrated a prolonged PT time. After further investigation, a deleterious mutation on the F7 gene was detected that was also passed onto his offspring [14]. Therefore, we know that factor VII deficiency is also present in Asian elephants, but the degree of its distribution in the population is unknown. Such knowledge is particularly important as Asian elephant can be struck by elephant endotheliotropic herpesvirus hemorrhagic disease (EEHV-HD), which causes acute generalized hemorrhagic diathesis due to capillary endothelial lesions [15]. How this vascular endothelial damage caused by the virus affects an elephant carrier of factor VII hereditary coagulopathy is still unknown; therefore, it is important to investigate. EEHV-HD have been intensely studied in the past two decades, especially in Asian elephants [15,16,17,18,19,20,21]. The disease is responsible for a high fatality rate in very young calves worldwide, reaching more than 50% of all captive born deaths above one day of life, for the US and European zoos [22,23,24]. Therefore, due to the impact of this disease and its hemorrhagic characteristics, research teams have dedicated more attention to the coagulation status of this species. Several coagulation assessment studies of Asian elephants have recently been published using different diagnostic methodologies based on human plasma as reference [14,25,26]. Additional studies focused on host blood viscoelasticity via thromboelastography [27,28,29]. Understanding elephant hemostasis has become a very important goal, both to improve the general knowledge base for elephant health status assessments, and to decipher the mechanisms by which EEHV-HD acts.

With this study we aim at increasing the knowledge of some of the most common coagulation parameters in a practical way and to obtain results in just a few minutes. Furthermore, we wanted to look at the possible genetic involvement of a hereditary factor VII deficiency on this disease onset and outcome. Although other coagulation factor deficiencies might be present, this investigation focuses only on the detection of a previously reported hereditary disorder involving factor VII in Asian elephants. For this, we have investigated the presence of mutations on the F7 gene in calves that have survived the disease, calves that have died with EEHV-HD, and other elephants that never presented symptomatology of EEHV-HD.

## 2. Materials and Methods

A total of 167 Asian elephants were assessed in our study. According to the specific parameter analyzed, the sample size varies, due to several reasons, because, for example, animals tested using stored frozen samples from past EEHV-HD fatalities were not assessed for coagulation times due to the impossibility to collect fresh blood. Fresh blood samples were collected from 127 Asian elephants in 21 zoos in Europe and in 10 Asian elephant touristic camps or farms in Thailand (Appendix A). Samples were collected by blood draw during routine check-up examinations, without sedation of the animals. Blood was either collected by natural flow, by using a butterfly needle or was drawn with a syringe attached to a needle with an adequate gauge size to avoid mechanical hemolysis. Most of the samples were collected by venipuncture of the ear vein, the rest were drawn from the saphenous vein in the hind leg.

### 2.1. Coagulation Time and Fibrinogen Measurements—Clinical Haemostasis Evaluations

After collection of the sample to a syringe, blood was distributed to a 2 mL ethylenediaminetetraacetic acid (EDTA) anticoagulating tube, and to sodium citrate tubes of 1.3 mL (3.2–3.8% concentration; KABE Labortechnik GmbH, Nümbrecht-Elsenroth, Germany), or, exceptionally, 2.5 mL citrate tubes were used. To avoid further hemolysis, the needle was detached before transferring the blood from the syringe to the tubes. Samples collected to EDTA tubes were stored in −20 °C or −80 °C. Specific sodium citrate-coated tubes were used immediately to measure the following three coagulation parameters: prothrombin time (PT), activated partial thromboplastin time (aPTT), and fibrinogen concentration. Whenever possible, PT and aPTT were performed immediately in the first 20–30 min after blood collection using the portable coagulation diagnostic analyzer VetScan^®^ VSpro Specialty Analyzer (ABAXIS Europe GmbH, Griesheim, Germany). Equipped with test specific cartridges, the analyzer allows in vitro determination of PT and aPTT times using cat and dog as reference species (Coagulation Cartridge, Abaxis Inc., Union City, CA, USA) and fibrinogen concentration (Fibrinogen Test Cartridge, Abaxis Inc., Union City, CA, USA) using horse as a validated species. The analyzer as also been used in smaller species and it is validated for lower volume of samples [30]; therefore, it is also applicable to use in small wildlife species. A combined PT/aPTT single test measurement offers a rapid quantitative result. A microcapillary designed test aspirates the citrated whole blood from a reservoir. By traveling through two parallel capillary paths, the blood is in contact with activators for coagulation. A light system detects when these microcapillaries blood flow stops, being this the test endpoint and the final quantitative coagulation time [31].

The fibrinogen test was measured by thrombin-mediated enzymatic conversion to fibrin, being applicable to other species [30].

All measurements were performed according to the manufacturer’s protocol. Analysis of fibrinogen were not always achieved for all individuals due to presence of hemolysis in the plasma samples. Cartridge loading with blood samples was performed very carefully to avoid hemolysis and foaming, both of which could lead to erroneous test results.

### 2.2. Platelet Counts

Blood smears were performed immediately after blood collection, using one drop of blood collected into the EDTA-coated tubes. These smears were then stained with Diff-Quick (Medion Diagnostics AG, Düdingen, Switzerland). The remaining EDTA blood was stored at −20 °C or −80 °C until further investigation. The stained smears were used to count the platelets under microscopic oil immersion objective observation. A total of ten fields were counted, and the final platelet count was obtained by calculating the average of these fields multiplied by 15,000.

### 2.3. Sample Collection for the Analysis of the Coagulation F7 Gene

For the analysis of coagulation F7 we used frozen blood samples collected into EDTA tubes and stored at −20 °C or −80 °C, in order to preserve DNA content. Tissue samples (liver, myocardium, tongue, etc.) from dead elephants were also analyzed, including samples from calves that died due to EEHV-HD (Appendix A).

### 2.4. DNA Extraction

For blood samples (200 µL EDTA-blood) from European animals, we used the “DNA blood extraction kit”, while for tissue samples, we applied the “Tissue DNA Mini extraction kit” (both peqLab Biotechnology, Erlangen, Germany). Thai Asian elephant DNA was extracted from blood samples, using Genomic DNA Mini Kit (Geneaid, New Taipei city, Taiwan). All extraction procedures followed the respective manufacturer protocols. DNA concentrations were measured using a NanoDrop^TM^ One/One^C^ spectrophotometer (Thermo Fisher Scientific, Waltham, MA, United States). DNA solutions were stored at −20 °C degrees.

All Asian elephant samples from European zoos were processed at the Leibniz-Institute for Zoo and Wildlife Research (IZW, Berlin, Germany) and analyzed for molecular characterization of the F7 gene. Asian elephant samples collected in Thailand were processed at the Faculty of Veterinary Science at Mahidol University, Bangkok, and sequenced externally (U2Bio, Sequencing Service, Bangkok, Thailand).

### 2.5. Amplification and Sequencing of DNA

Primer sequence information and amplification conditions were obtained from a previous study [14], and optimized with minor modifications to cycle number and primer combination (Table 1).

PCRs were performed in a final volume of 25 µL consisting of 2 µL of DNA extract, 0.5 µL (final concentration of 0.2 µM) of each primer (Biolegio, Nijmegen, The Netherlands), 12.5 µL DreamTaq MasterMix, and completed with 9.5 µL nuclease-free water (both Thermo Scientific, Vilnius, Lithuania).

PCRs were performed on G-STORM GS1 thermocycler (Gene Technologies Ltd., Somerton, UK). Cycling conditions for all exons but exon 4 were: 95 °C 3 min, 35 × (95 °C 30 s, 58 °C 30 s, 72 °C 1 min), final extension at 72 °C 7 min, followed by eternal 20 °C. For exon 4 we applied an annealing temperature of 53 °C. Presence of PCR products was visualized by electrophoresis on 1% agarose gels.

Prior to the subsequent sequencing excess primers and dNTPs were removed using the ExoFastAP Purification Kit (Thermo Scientific^TM^, Schwerte Germany). Amplified F7 exons were then Sanger sequenced bidirectionally using the BigDye Terminator Cycle Sequencing kit (Applied Biosystems – Thermo Fisher Scientific, Waltham, MA, USA). Terminated fragments were separated on an ABI 3130*xl* Genetic Analyzer and visualized using the Sequencing Analysis Software v5.2 (both Applied Biosystems^®^, USA). After removal of primer sequences F7 exon fragments were mapped to using Geneious (v8.0.5, https://www.geneious.com, San Diego, CA, USA), and *Loxodonta africana* F7 was used as a comparative reference (Genbank acc. no. NM_001330481.1). Finally, we used the software SeqMan Pro (DNASTAR Lasergene package v11.2.1, Madison, WI, USA) to generate the assembly of the gene.

Amplifications and sequencing were performed for 65 European elephants (from a total captive population of 307 individuals [32]). Only six of the Thailand population had the full gene sequenced. The other 70 were only sequenced for the exons which we determined to present missense (non-synonymous) mutations (exons 2, 4, and 5), due to cost restriction. Nevertheless, the samples obtained in Thailand were amplified using the same PCR protocol and same primer pairs. Successful amplicons were sent to U2Bio (Bangkok, Thailand) Sequencing Service for DNA sequencing, and the obtained sequences were sent to IZW, Berlin, to be added to the Asian elephant F7 data set for analysis in Geneious (v8.05).

### 2.6. Data Selection and Analysis

Average coagulation times were estimated per study region (Thailand, Europe) and then estimated by age class for each gender. We assigned animals to 1 of 5 age classes: class (1) from birth until four years old, class (2) from five to nine years old, class (3) from ten to 19 years old, class (4) from 20 to 34 years old, and class (5) older than 35 years. Two fetuses (one male and one of unknown gender) were sequenced and analyzed for the F7 gene and belong to age class 0; therefore, these are not represented in the coagulation parameters analyzed with fresh blood. The category of EEHV-HD status separated healthy individuals that never presented EEHV symptoms and calves which were diseased with EEHV-HD. Database can be found in Appendix A).

Univariate analysis of variance (UNIANOVA) and tests of between-subject effects were used to evaluate the effects of the study region and EEHV-HD status on overall coagulation times, fibrinogen concentration, and platelet count values. The same statistical tests were used to investigate the influence of gender and age class in the mean results of PT and aPTT times, fibrinogen concentration, and platelet counts. To account for multiple comparisons, we performed multiple-comparison post hoc statistical tests (Tukey-HSD and Bonferroni).

Single nucleotide polymorphisms (SNPs) were evaluated for possible impact on the factor VII protein structure in comparison with complete F7 gene, which, currently, is only available for *Loxodonta africana*, as carried out by the reference study [14], and using the web-based protein variation effect predicting software packages SIFT [33] and PROVEAN v1.1.3 [34]. For the mutations considered to be deleterious and not tolerated, a Kruskal–Wallis test was applied to evaluate the impact of the SNP in the PT time of coagulation. To compare the genotypes, more specifically, to assess the differences between SNPs causing missense mutation, between region (Thailand and Europe) and between different EEHV-HD status (regardless of the region), we used Fisher exact test and chi-square tests.

Data analysis using UNIANOVA, tests of the between-subject effects and post hoc multi comparison tests were conducted using IBM SPSS Statistics (v24-COMPLETAR) predictive analytics software. Missense mutation analysis using Fisher’s exact test, chi-square tests and drawing of graphs were performed using GraphPad Prism (v9, GraphPad Software, San Diego, CA, USA). Statistical significance was designated at *p* ≤ 0.05. Unless stated otherwise, results in the text are presented as means.

## 3. Results

### 3.1. Overview of the Study Population

A total of 167 Asian elephants (*n* = 76 Thai elephants and *n* = 91 European), were analyzed. Females (*n* = 104) were on average 25 years old (SD 14), while males (*n* = 37) were on average 18 years old (SD 16). This age difference was significant (*p* = 0.013) for both regions, but not significantly different within regions (*p* = 0.149).

For the Thai population, we found that both females and males were, on average, 21 years of age. On the other hand, European female elephants in the study presented a mean age of 28 years old, while the males were younger, at around 17 years of age.

### 3.2. Influence of Location and EEHV-HD Status on Coagulation Time, Fibrinogen Concentration, and Platelet Counts

Results presenting means, SD, and population sample size used for each of the following tested parameters are presented in Table 2.

#### 3.2.1. Coagulation Times

A total of 127 elephants (63 from Thailand and 64 from Europe) were assessed for their prothrombin (PT) and activated partial thromboplastin time (aPTT). PT time was significantly lower in the Thai group then in the European group (*p* = 0.026). Thai elephants had an average PT of 17.16 ± 0.58 s, while European elephants had an average of 17.55 ± 1.23 s. In contrast average aPTT was significantly lower (*p* < 0.0001) in the European group (125.57 ± 17.30 s), than in the Thai group (141.86 ± 19.77 s; Figure 1).

Concerning EEHV-HD, no significant difference in PT time (*p* = 0.158) was found between the group of calves having survived the disease (*n* = 8) and the rest of the population (*n* = 119). In contrast, aPTT was significantly different between these calves and the rest of the population (*p =* 0.004).

#### 3.2.2. Fibrinogen

One individual from Europe was removed as outlier from the comparison due to its extremely high fibrinogen concentration (1633 mg/dL). Therefore, we investigated fibrinogen concentration in a total of 117 elephants and a highly significant difference between the two groups was found (*p* < 0.0001). The Thai elephant group (*n* = 63, had a lower mean value (468 ± 116 mg/dL), than the European group (*n* = 54, 601 ± 111 mg/dL). Fibrinogen values of calves that had survived EEHV-HD (*n* = 7) did not significantly differ (*p* = 0.902) from animals that never had the disease (Figure 1).

#### 3.2.3. Platelet Counts

Although Asian elephants from European zoos had, on average, higher overall platelet counts, that difference was not significant (*p = 0.*376) in comparison to the Thai elephants and a minimal difference on the platelet counts separate them; Europe: *n* = 60, mean = 594 × 10^3^/μL ± 182; Thailand: *n* = 55, mean = 558 × 10^3^/μL ± 271. Platelet counts found for EEHV-HD survival calves were also not significant (*p* = 0.708) (Figure 1).

### 3.3. Influence of ender and ge lass on Coagulation Time, Fibrinogen Concentration, and Platelet Counts

Although not significant, males had a slightly prolonged coagulation time, a lower platelet count, and lower fibrinogen concentration than females (Table 2).

#### 3.3.1. Coagulation Times

No effects associated with age (*p* = 0.816) or sex (*p* = 0.700) were found to influence the overall prothrombin time. An overall average of 17.36 s was found (±1.10) for the Asian elephants in the study (Table 2).

In similarity with the results of PT times, we found that for aPPT no age (*p* = 0.442) or gender (*p* = 0.504) was associated with difference in values, and on average, we found that this coagulation route lasts 131.60 s (±18.45) (Table 2).

#### 3.3.2. Fibrinogen

For the 103 elephants analyzed for fibrinogen concentration (females *n* = 81, males *n* = 22), an average of 561 mg/dl (±166) was found (Table 2). Sex did not prove to have an impact on the total fibrinogen concentration (*p* = 0.419). However, although not significant, the first two age groups—(1) calves until four years of age and (2) young elephants from five to nine– years old—presented an overall lower value of fibrinogen, compared with elephants of older ages, independent of the region. Furthermore, between the age group (2) (from -9 years old) and group (5) (older than 35 years), there was a borderline effect for the younger ages to present a lower average fibrinogen concentration (*p* = 0.057).

#### 3.3.3. Platelets

No significant difference on the platelet counts was recorded between the different age groups (*p* = 0.231), or gender (*p* = 0.665), and an average of 572 × 10^3^/μL was found for the elephants in our study (Table 2).

### 3.4. Coagulation Factor VII Gene (F7)

#### 3.4.1. Analysis of F7 Gene Sequences

The alignment of sequences from 141 individuals were compared with the available *Loxodonta africana* reference and showed ten polymorphic positions (Table 3) distributed in exons 2, 4, 5, and 8. Of these, six were silent (synonymous), but four caused missense (non-synonymous) mutations. These SNPs were present in exons 2 (C193G), 4 (C332T), and 5 (T437A and G509A).

#### 3.4.2. Distribution of Missense SNPs in the European and Thai Populations

The distribution of polymorphisms causing missense mutations was significantly different between study regions only in exon 2 (*p* < 0.0001), exon 4 (*p* = 0.47), and exon 5 (T386A *p* = 0.59 and G458A *p* = 0.89; positions of the SNPs detected in the F7 gene can be found in the Appendix A).

For the biallelic SNP C142G on exon 2, we sequenced 136 animals of which 80 were homozygous for this SNP (C/C *n* = 67, G/G *n* = 13) and 56 were heterozygous. When comparing exon 2 SNP allele distribution by study region, we found a significant difference between the regions. In the Thai elephant population analyzed (*n* = 76), the majority of individuals (66%) carried the homozygous C/C wild-type genotype (i.e., the allele from the *Loxodonta africana* reference sequence), while among the Asian elephants analyzed from European zoos (*n* = 60), the majority (57%) carried the heterozygous C/G genotype (*n* = 34) (Table 4). The exon 2 “G”-allele of the F7 gene will cause a substitution of leucine by Valine (Leu48Val) in the factor VII protein. Both protein effect prediction software packages (SIFT, PROVEAN) considered such amino acid exchange to be tolerable and likely to have a neutral effect (Table 4).

There were two of the three SNPs detected in F7 exon 4 which were synonymous, while the 3rd (C281T) was non-synonymous. The exon 4 “T”-allele causes an amino acid substitution from proline to leucine (Pro94Leu; Table 4). The majority of both Thai elephants (*n* = 70) and of Asian elephants from European zoos (*n* = 49) were homozygous for the wild-type “C”-allele encoding proline at that position. A total of eight animals (Thailand *n* = 6, European zoos *n* = 2) were heterozygous and none were homozygous for the “T”-allele. The substitution of proline by leucine was predicted to be deleterious for the structural integrity of factor VII protein (Table 4). Out of the six heterozygous individuals that had been tested for PT time, five had a significantly higher PT time than the population mean (*p* = 0.017).

In F7 exon 5, we detected one silent mutation and two missense mutations. For the first one of the two missense mutations, no animal was homozygous for the wild-type (*Loxodonta africana*) “T”-allele and only three elephants were heterozygous (T/A). The majority of the population (across both “study regions”) was homozygous for the “A”-allele. This allele leads to a substitution of leucine by glutamine (T386A; Leu129Gln), which was predicted to be deleterious for protein integrity. Heterozygous elephants “A/T” were very rare (*n* = 3). For the second SNP having a missense allele (G458A), we only detected three animals to carry the non-synonymous “A”-allele (in a total of 141 sequenced individuals). One of them was homozygous (A/A) and two heterozygous (G/A). The “A”-allele will cause a substitution of arginine by glutamine (Arg153Gln). However, this alteration was predicted to be tolerated and neutral (Table 4).

#### 3.4.3. Distribution of Missense SNPs between non-EEHV and EEHV Symptomatic Cases

We found no association between the distribution of any of the detected missense SNPs, neither being heterozygous nor homozygous, and a previous symptomatology of EEHV-HD. Thus, none of the missense mutations detected in this study could be associated with the chance of developing EEHV-HD (*p* > 0.05, for all exons).

## 4. Discussion

In the present study, we analyzed the coagulation time (PT and aPTT), fibrinogen concentrations, and platelet count of Asian elephants from 10 camps in Thailand and 21 European zoos with a new and fast results method. The large dataset presented here, which gathered data from a broad range of age classes, gives us good reference values for coagulation parameters in the Asian elephant population. To the best of our knowledge, this is the first study using a VSPro, a very fast diagnostic point-of-care analyzer, specifically for measuring coagulation time and fibrinogen concentration in elephants. Furthermore, we investigated the presence of genetic mutations in the coagulation F7 gene, and their possible connection to hereditary coagulation disorder.

### 4.1. Fast Diagnostic Analyzer (VSPro)

The materials and methods used in this study to obtain coagulation times and fibrinogen concentration were designed to minimize procedure times. The VSPro analyzer has been previously used and compared with other traditional laboratory methods in dogs [23], where it yielded reliable results for detecting abnormalities in PT and aPTT [35]. Similar to these previous results, in the present study, we obtained a readout of a PT/aPTT coagulation time as fast as 3 min. Together with its simplicity of use, the VSPro analyzer proved to be of advantage when analyzing these parameters in elephants.

The fastest result for fibrinogen concentration obtained in our study was at 10 min after the initiation of the protocol. Therefore, our method is so far the fastest technique for PT/aPTT measurement and fibrinogen concentration evaluation in elephants, which will be essential in emergency situations, as for example during an EEHV-HD outbreak. Our approach also allowed us to drastically reduce the possibility of laboratory work associated errors and divergences that would derive from sending the blood samples for analysis to different laboratories, potentially even using different methods. The manufacturer’s protocol recommends the equipment to be used between 15 °C and 30 °C. These recommendations could not always be followed during fieldwork, as it was necessary to analyze some samples at temperatures < 15 °C (winter in Europe) and > 30 °C (summer in Thailand). There was only one day with ambient temperatures > 40 °C when the VSpro stopped functioning properly (displaying an overheating alert). On all other out-of-recommended-range temperature days the device worked normally.

Venipuncture and blood drawing are part of a routine veterinary procedure to check an animal’s health status. It is also considered a minimally invasive method for sample collection. Therefore, when combined with regular check-ups, no additional stress or pain was caused to the animals sampled for this study. Additionally, this procedure did not require sedation, which could affect the blood coagulation cascade. Low stress is also important to prevent the risk of spleen contraction, which could increase cell count, platelet count, and aggregation, and alter several coagulation factors levels, including fibrinogen. This would influence the coagulation cascade and bias the PT and aPTT measured in this study.

In most extinction-threatened species, coagulation is still rarely investigated. We hope with this research to provide practitioners and researchers with a quick and simple tool that can be easily implemented to further explore coagulation research in zoo and wildlife species.

### 4.2. Coagulation Times

We found that neither gender and nor age influenced PT and aPTT times, fibrinogen concentrations, and platelet counts. Although the difference was not significant, males had in general a prolonged coagulation time, a lower platelet count, and a lower fibrinogen concentration than females. Unfortunately, our data set consisted of almost three times as many females than males (85 vs. 27). So, this result needs to be further investigated in a better gender-balanced study.

A significant difference, however, was found in PT times between Asian elephants from European zoos and Thai camp elephants. Due to our large sample size, we were able to even detect a small difference of just 0.39 s. However, such small difference will not cause any biological effect of clinical significance in the coagulation capacity, and therefore no therapy is advised. Our PT results (*n* = 127, mean = 17.36 s, SD = 1.07 s) were higher than previous studies reported for Asian elephants using much smaller sample sizes and different measuring methods (*n* = 7, PT-simp: mean = 9.6 s, SD = 0.7 s and PT-inn: mean = 10.3 s, SD = 1.1 s [25]; *n* = 6, PT median = 14.74 s (range 11.6–20.9 s) [26]; *n* = 23, PT: median = 11.0 s (range 9.7–14.9 s) [29]). This suggests that each measuring method will give different results for healthy elephants rendering the comparison of results from different methods impossible without reference samples.

For aPTT, we also found a statistically significant difference between elephants from the two study regions, with lower times for the European zoo elephant population. Because standard deviations in both gender groups and in each age class analyzed was higher than the difference found between study regions, we do not expect this result to translate into biological effects in the coagulation capacity. Nevertheless, we found a big range of aPTT values in our study population (*n* = 127; min = 84 s, max = 194 s) when compared with other species also analyzed with VSPro, where a smaller range of time is reported (manufacturer reference ranges: dogs 71–105 s, cats 86–137 [2], other studies: *n* = 109 guinea-pigs 61–84 s [30], *n* = 14 wallabies 71–84; although, PT could not properly be measured using this POC for this species [36]). In a normal human population, aPTT is known to also vary greatly between individuals, and this wide reference range interval is affected by several causes, such as biological variability, instrumentation and reagent variability, and physiological changes, such as pregnancy, physical stress, or trauma [37].

Previous tests to determine aPTT values in elephants used different methods and were based on human plasma as reference for comparison [25,26,29]. Their results differ greatly in scale from the values measured in our study. A coagulation deficiency is reported to become evident when PT or aPTT is greater than 1.5× above the upper end of the reference range [30]. Assuming that this applies as well for other mammalian species and specifically to elephants, this supports the notion that the PT and aPTT differences found between “study regions”, even though they were statistically significant, do not bear clinical relevance. Accordingly, and combined with the healthy status of the animals sampled and all values being within the manufacturers range limit, we assume that all animals in our study had a normal aPTT coagulation time. The combined results from PT and aPTT measurements suggest that these values should define a new reference value for practitioners using this method on Asian elephants and will allow to stop hitherto applied comparisons with several different techniques. Having a large data set composed of Asian elephants from different regions and from a wide age range, we consider our values reliable and reproducible.

Regarding the findings on the EEHV-HD status, no significant difference was found for PT time between the surviving calves and the rest of the population (*p* = 0.158). In contrast, aPTT varied significantly between these groups (*p =* 0.004). However, both results could be due to the difference between sample sizes (*n* = 8 survivors; *n* = 119 non EEHV-HD cases), where the number of survivors might be too small to detect a difference.

### 4.3. Fibrinogen

Although not significant, we found a tendency for the younger age classes (0–9 years of age) to have a lower fibrinogen concentration than the older elephants (especially those older than 35 years), independent of the study region.

In our Asian elephant sample set the mean fibrinogen concentration was 561 ± 166 mg/dl, higher than reported in previous studies [25,26,29,38,39]. As these studies had used measuring methods differing from ours, we assume that this may be a method related difference.

We found an outlier in our initial study population, presenting more than three times the average concentration found for the sampled population in the study. This animal was sampled during the process of fetal mummification. Fetal retention in elephants is not an uncommon phenomenon and there are several reports of interrupted parturition with retention of up to 84 months [40,41,42]. This finding emphasizes the importance of fibrinogen measurement, which can be used as a useful diagnostic tool for health routine check-up.

### 4.4. Platelets

An average of 572 × 10^3^ platelets/µL was found for the elephants in our study, which is in accordance with previous Asian elephant hematology studies [38,43,44] and lower than one reported study [45]. Several outliers were found in the platelet count analysis, presenting values reaching up to 1343 × 10^3^ platelets/µL. No disease was diagnosed at the time of sampling for these animals, so we cannot attribute these results to any sickness or health compromised status.

### 4.5. Genetic Analysis of F7 Gene

A previous study in the F7 gene of Asian elephants reported a deleterious mutation in a single nucleotide position (SNP A202G) which was attributed to prolong PT time [14]. Although the animals investigated in our study did not carry this SNP, we found ten new point mutations—six were considered to be synonymous or silent, and four non-synonymous or missense. Two of these missense mutations were predicted by SIFT and PROVEAN to be tolerated or to have a neutral impact in the protein structure. The other two non-synonymous variants correspond to Pro94Leu (exon 4, C281T) and Leu129Gln (exon 5, T386A), and they were both predicted to be not tolerated and to cause deleterious changes in the protein. Proline has a cyclic structure and since it is the wild-type protein, we assume that there is a bending in the structure of factor VII at that location. According to SIFT predictions, Proline cannot be substituted by any other aa. Therefore, having a leucine (aliphatic and open chain structured) at that point would alter the protein structure and invalidate its coagulation functioning. However, there were no homozygous individuals in our study with the mutant type, meaning all Asian elephants have at least one wild-type functioning allele. From the six heterozygous elephants with this variant, five had higher coagulation PT times (which is influenced by factor VII activity) than the average in the study. However, these individuals had only a prolongation of nearly one second and have a lower mean that the upper quadrant. Therefore, no reliable conclusion on their predisposition to have a coagulation deficiency can be made and they were considered healthy.

In exon 5, at position T386A, the F7 gene from *Loxodonta africana* (used as reference here) has a “T”, the triplet thus coding for a Leucin. None of the *Elephas maximus* present in the study presented this homozygous wild-type nucleotide. Our Asian elephant population is 97% homozygous with a mutant-type allele (A/A), causing a shift to glycine, and only three individuals were heterozygous (A/T). The change from leucine (non-polar and hydrophobic aa) to glycine (polar and hydrophilic aa) was predicted as deleterious. We predict that this amino acid resides in a position of the protein, which is not actively involved in the coagulation process, because PT times for animals that were homozygous or heterozygous for this mutation were with the normal range.

Although factor VII deficiency is a rare disease and more than 200 genetic variants have been reported in humans so far [11,12], some of these mutations seem to be recurrent and a few with relatively high frequency [10]. We found only one SNP to be significantly different between the two regions (exon 2, C142G, Leu48Val), with the majority of the Asian elephant Thai population being homozygous wild-type and the majority of the Asian elephants from European zoos heterozygous for the mutation. This significant distribution of genetic variance was not accompanied by a difference in PT times. This mutation was also considered as tolerable or neutral by the predicting software, therefore we cannot assume that the integrity of factor VII and consequently the extrinsic coagulation pathway will be affected by the presence of this SNP.

## 5. Conclusions

Knowing the physiological status of the coagulation of Asian elephants is of great importance, as it provides a baseline of normal ranges to compare with, when facing diseased situations, such as an EEHV-HD outbreak. This study was performed in Asian elephants living in Thailand and in Europe and it give us a reference range of normal values of coagulation parameters—PT, aPTT times and fibrinogen concentrations—discriminated between different age groups, genders, and regions. Samples were, for the first time, processed using a very practical point-of-care analyzer (VSPro) and most results were achieved under 20 min, making it a suitable diagnostic method for emergency cases, and in the field of Asian elephant range countries.

Although we have found significant differences in the coagulation times between the European and Thailand populations, the time gaps reported were very low; therefore, they were not expected to cause any biological effect.

With this study, we have improved the knowledge of F7 gene variation in Asian elephants. We found ten intraspecies variations that can be used as reference for future F7 gene analysis in Asian elephants. Findings on coagulation F7 gene revealed several single nucleotide position mutations in the population that did not translate to a significant alteration in the coagulation time of the individuals with the mutations.

Due to lack of financial, time, logistical, and human resources, it was not possible to run all the validation tests during this investigation. However, as a future perspective, we believe it would be an important topic for further research. Nevertheless, the large sample size used in this study and the results obtained are good indicators that this POC can be used in Asian elephants. These preliminary results are important for future clinical practice comparisons.

## Figures and Tables

**Figure 1 animals-12-00361-f001:**
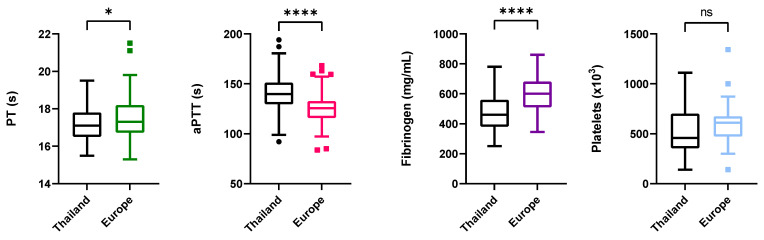
Boxplot representation of the PT, aPTT, fibrinogen, and platelets values grouped by study region. The box represents the 25th–75th percentile values of the distribution (interquartile range), the line within the box represents the median (50th percentile), and the whiskers approximate the 2.5th and 97.5th percentile values. Stars indicate significance threshold. *: *p* < 0.05, ****: *p* < 0.0001; ns—not significant.

**Table 1 animals-12-00361-t001:** Names and sequences of the forward and reverse primers (5′–3′) used to amplify the eight exons of the F7 gene.

F7e1_F	GAGCAGCTGAGGAACTTAGC	F7e1_R	CCCACTTTCCAGATTTGAGG
F7e2_F	TACAAGCCAGGAGAAGGAGC	F7e2_R	ATGGACTCCAGGAGACATGG
F7e3_F	TCTGTGGCTGACTTGTTTGC	F7e3_R	AGAAGGGGGTGAGGTAGGG
F7e4_F	AACTCACCGCCATCTCTCC	F7e4_R1	TCAACACTCTCAGATTGGAAGG
F7e5_F	CTGTACCAGCTGCTTTTCCC	F7e5_R1	TCAGTAAAGGTTATGCCCGC
F7e6_F	AGCTCAGGCAGATGTAACCC	F7e6_R1	GCTGACCTGCCATTTTTCTC
F7e7_F	GCCAGATAAGAGGGCAGTTG	F7e7_R1	CGATAGCAGAGAGGTTTGCC
F7e8_F1	TGACAGGCCAAAGACACAAC	F7e8_R1	GTCCCATCCAGGTAGCCAG
F7e8_F2	ACGTAGTGCCCCTCTGTTTG	F7e8_R2	GCAGCAGCAGCTTTATTTCC
F7e8_F3	TCTCCCGGTACATTGAGTGG	F7e8_R3	GACGTCCATCTCTCTCAGCC

In [11], exons 3 and 4 were amplified with two different primer pairs. From these, we only used the 2nd pair for exon 3 and from exon 4 information we used the forward primer 1 in combination with reverse primer 2 to amplify the final exon 4.

**Table 2 animals-12-00361-t002:** Estimated mean and SD of prothrombin time (PT), activated partial thromboplastin time (aPTT), fibrinogen, and platelet counts for different groups of individuals, sorted according to study region, gender, age class, and known presence or absence of EEHV-HD.

		PT (s)			aPTT (s)			Fibrinogen (mg/dL)			Platelet Count (×10^3^/μL)		
EEHV	REGION	Mean	SD	N	Mean	SD	N	Mean	SD	N	Mean	SD	N
No EEHV-HD	Thailand	17.13	0.86	57	143.80	18.68	57	467	112	57	540	274	49
	Europe	17.51	1.23	62	126.10	17.31	62	601	179	54	604	173	58
	Total	17.33	1.08	119	134.58	19.98	119	530	167	111	575	226	107
EEHV-HD survivors	Thailand	17.45	0.54	6	123.40	22.14	6	481	162	6	701	218	6
	Europe	18.70	0.71	2	109.15	2.90	2	560	.	1	281	198	2
	Total	17.76	0.79	8	119.84	19.87	8	492	151	7	596	278	8
Total between Groups	Thailand	17.16	0.84	63	141.86	19.77	63	468	116	63	558	271	55
	Europe	17.55	1.23	64	125.57	17.30	64	601	111	54 *	594	182	60
	Total	17.36	1.07	127	133.65	20.22	127	530	132	117	576	229	115
**Gender**	**AGE class**	**Mean**	**SD**	**N**	**Mean**	**SD**	**N**	**Mean**	**SD**	**N**	**Mean**	**SD**	**N**
F	1	16.88	0.74	4	128.43	2.08	4	526	153	4	433	215	3
	2	17.12	1.40	9	125.61	21.56	9	423	103	9	693	408	6
	3	17.46	0.91	11	133.43	16.45	11	555	135	11	500	167	10
	4	17.46	1.13	37	132.18	18.67	37	560	116	35	627	216	36
	5	17.26	1.14	24	130.42	17.20	24	661	247	22	522	181	24
	Total	17.34	1.11	85	130.97	17.66	85	570	176	81	577	225	79
M	1	18.00	1.03	4	118.25	10.66	4	597	35	3	555	336	4
	2	16.98	0.43	4	133.05	17.02	4	578	119	3	615	58	4
	3	17.38	0.68	9	141.07	15.36	9	503	140	8	646	249	7
	4	17.53	1.99	6	130.13	35.49	6	533	57	4	448	116	4
	5	17.28	0.51	4	137.90	12.53	4	470	162	4	394	210	3
	Total	17.43	1.08	27	133.60	20.99	27	525	119	22	553	223	22
Total between “age classes”	1	17.44	1.03	8	123.34	8.95	8	556	116	7	503	276	7
	2	17.08	1.17	13	127.90	19.88	13	462	123	12	662	308	10
	3	17.43	0.80	20	136.87	16.03	20	533	136	19	560	211	17
	4	17.47	1.25	43	131.89	21.19	43	558	111	39	609	215	40
	5	17.27	1.07	28	131.49	16.63	28	632	244	26	508	185	27
	Total	17.36	1.10	112	131.60	18.45	112	561	166	103	572	223	101

Age classes were: class (1) 0–4 years old, class (2) 5–9 years old, class (3) 10–19 years old, class (4) 20–34 years old, and class (5) > 35 years old. F—female; M—male; SD—standard deviation of mean; N—number of individuals in the respective group. * After removal of outlier animal “122” (Appendix A). EEHV-HD—elephant endotheliotropic herpesvirus hemorrhagic disease.

**Table 3 animals-12-00361-t003:** Single nucleotide polymorphisms (SNPs) found in four exons of the coagulation factor F7 gene in the Asian elephants evaluated in this study. SNPs are listed according to their position, alteration in the codon, and type of mutation.

Exon	SNP Position	Codon Change	Type of Mutation
2	C142G	CTG > GTG	missense
4	C281T	CCG > CTG	missense
	G294C	GGG >GGC	silent
	G300C	CTG > CTC	silent
5	T386A	CTG > CAG	missense
	G458A	CGA > CAA	missense
	T489C	GAT > GAC	silent
8	C870T	CGC > CGT	silent
	C975T	AGC > AGT	silent
	T1161C	AGT > AGC	silent

Positions refer to the *Loxodonta africana* F7 cDNA without 5‘untranstlated region (Genbank acc.no NM_001330481.1). The actual position of the SNP in the triplet is underlined.

**Table 4 animals-12-00361-t004:** Distribution of the missense mutations found, by region, by non-EEHV symptomatic elephants and EEHV-HD symptomatic calves. Prediction of aa substitution and its impact on the biological function of the protein tested are presented for both PROVEAN and SIFT software.

Missense SNP	Amino Acid	SIFT	PROVEAN	State	Thailand	Europe	Total	No EEHV-HD	EEHV-HD	Total
exon2, C142G										
C				wild-type *	50	17	67	62	5	67
C/G	Leu48Val	Tolerated	Neutral	Heterozygous	22	34	56	50	6	56
G	Leu48Val			Homozygous different	4	9	13	11	2	13
Total					76	60	136	123	13	136
exon4, C281T										
C				wild-type	70	49	119	108	11	119
C/T	Pro94Leu	Not Tolerated	Deleterious	Heterozygous	6	2	8	7	1	8
Total					76	51	127	115	12	127
exon5, T386A										
T				wild-type	0	0	0	0	0	0
A	Leu129Gln			Homozygous	75	63	138	126	12	138
A/T	Leu129Gln	Not Tolerated	Deleterious	Heterozygous	1	2	3	3	0	3
Total					76	65	141	129	12	117
exon5, G458A										
G				wild-type	75	63	138	126	12	138
G/A	Arg153Gln	Tolerated	Neutral	Heterozygous	0	2	2	2	0	2
A	Arg153Gln	Tolerated	Neutral	Homozygous	1	0	1	1	0	1
Total					76	65	141	129	12	141

* wild-type here indicates matching to the F7 gene of *Loxodonta africana* (genebank acc.no. NM_001330481.1); EEHV: elephant endotheliotropic herpes virus; EEHV-HD: EEHV hemorrhagic disease.

## Data Availability

The data presented in this study are available in the Appendix A.

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
