# Peer review of "Assessing Coagulation Parameters in Healthy Asian Elephants (Elephas maximus) from European and Thai Populations"

_animals, 2022, doi:10.3390/ani12030361_

Round 1

Reviewer 1 Report

Thank you for working on a study on this important topic in elephants. I agree that further investigation into elephant coagulation is needed. I believe that several items need to be addressed in this research paper, however, prior to publication.

The main overall concern I have is the lack of validation in using Abaxis system for analyzing elephant blood. I don't believe that your significant differences between your populations are caused by differences in your populations--instead I believe it is simply due to different locations or different analyzers. I strongly suggest that you show that the analyzer gives similar results. Ideally, you would have run each elephant's blood sample three times and made sure you received results within 10% of one another or something similar, but, with this data already collected, a small study of 5-30 elephants where repeatability of the results would be advisable. It would also be recommended to dilute out some of the samples and see if you can receive a linear result and also determine if your PT and aPTT could give a linear response when mixed with calcium, kaolin, or another activator. Some sort of validation is needed with this study. Perhaps even running PT/PTT/fibrinogen with other methods concurrently and seeing if the results align. Far too much stock is put into the results of this analyzer with no validation. Also, you cannot accept results when the analyzer was run at temperatures outside of Abaxis' guidelines, especially if no validation was performed to show that the results should not have been affected. Perhaps you can run a single elephants sample in a warm room and at room temperature and a cold room at one of your research areas? Additionally, PCV is known to be highly influential on coagulation parameters, so please report this for all animals included in the study.

Additional notes:

  • The introduction discussing factor VII deficiencies doesn't specify that it is discussing it in humans at first and it is very confusing.
  • The introduction is quite clunky. I recommend reworking it and presenting it as you are trying to classify as much as you can about the coagulation cascade in Asian elephants.
  • Please justify comparing the Asian elephant factor VII gene sequence to that of African elephants. It is very likely the correct comparison, but it should be justified
  • All of your insistence that coagulation testing should be done routinely with blood work and receiving more immediate coagulation test results when treating EEHV are stated without any support. How would having this information on a routine basis be helpful? How would having these results sooner help--would clinicians start plasma treatment sooner? Would it help with prognosis?
  • It seems concerning that calves recovered from EEHV would still have significantly different aPTT values. Do you actually believe this is the case, or do you believe this is spurious. Do you think there are long lasting coagulation effects of this disease process?
  • I understand how in field situations performing an estimated platelet count on a slide is what needs to occur, but I would specify that you are performing an estimated platelet count and I would delve more into the methods that hopefully made this a very standardized process--did you invert the tube a certain amount of times prior to withdrawing the blood for the slide? Did you check the feathered edge for platelet clumps and determine that the slides were free of them?

Author Response

Dear reviewers,

We are very thankful for your review and comments on our manuscript. We have conducted extensive revisions in agreement with your inputs and we believe the manuscript has been greatly improved. The line numbers referenced below correspond to the “simple markup” option, inside the tab “Review”> Track changes. You can follow all details of the changes made to the manuscript if you select the “all markup” option in the same tab. We thank you greatly for your time, corrections, and suggestions.

Please find the replies to your comments below.

Reviewer 1

Thank you for working on a study on this important topic in elephants. I agree that further investigation into elephant coagulation is needed. I believe that several items need to be addressed in this research paper, however, prior to publication.

Point 1: The main overall concern I have is the lack of validation in using Abaxis system for analyzing elephant blood. I don't believe that your significant differences between your populations are caused by differences in your populations--instead I believe it is simply due to different locations or different analyzers. I strongly suggest that you show that the analyzer gives similar results.

Response 1: Thank you for this important comment. Although we found statistically significant differences with the statistical test used, these differences refer to 0,39 s (PT) and 16,29 s (aPTT). A coagulation deficiency is reported to become evident when PT or aPTT is greater than 1.5× of the maximum value of the reference range. Assuming that this applies as well to elephants, this supports the notion that, even though they were statistically significant, the PT and aPTT differences found between ‘study regions’, don’t bear clinical biological relevance.

The analyser used was the same in both regions (the exact same machine).

For a normal human population, aPTT is known to vary greatly between individuals, and a wide reference range interval is affected by several causes such as biological variability, like for example pregnancy, physical stress, or trauma. It is quite possible that some of the females were pregnant at the time of sampling since breeding is encouraged in this critically endangered species. Nevertheless, we believe this difference is not relevant enough to project a biological alteration in the coagulation capacity, but more a statistical finding.

We do agree that results should be taken carefully, and we must take in consideration that for example in the PT we had 2 outliers that will influence the overall results by increasing the final mean. We do not know what other factors might be related and not being considered and there is always the possibility of this results being a false positive (type I statistical error). This puts in evidence that this topic should take more future attention and that further research would be important, as for example regular measurements to assess intraindividual and interindividual differences in populations.

Nevertheless, and again due to the small differences found between groups, we believe this study to be of great importance for future comparisons of coagulation time in this species.

Point 2: Ideally, you would have run each elephant's blood sample three times and made sure you received results within 10% of one another or something similar, but, with this data already collected, a small study of 5-30 elephants where repeatability of the results would be advisable.

Response 2: Thank you for your comment. Repetitions were made as following:

1) Three animals with 3 sample measurements repeated for PT and aPTT,

2) Five animals with 2 sample measurements repeated for PT and aPTT,

3) Eleven animals with 2 sample measurements repeated for Fibrinogen.

We have defined the parameter of reproducibility as excellent for a coefficient of variation (CV) ≤10%, good for CV between 10–20%, acceptable for CV between 20–30%, and poor for CV >30%. Our results were all excellent (<10% CV) for PT and excellent or good (10-20%) for aPTT and fibrinogen.  Overall, we got all coefficients of variation under 17%, which is considered good, although for a very small sample size.

Europe

CV% PT

Score

CV% aPTT

Score

Europe

CV% Fibrinogen

1

0.4

excellent

0.4

excellent

1

11.4

good

2

1.1

excellent

1.0

excellent

2

5.6

excellent

3

1.2

excellent

1.6

excellent

3

6.1

excellent

4

1.7

excellent

8.9

excellent

4

7.6

excellent

5

1.7

excellent

11.0

good

5

3.1

excellent

6

2.7

excellent

14.0

good

6

0.0

excellent

7

3.9

excellent

16.8

good

Thailand

Thailand

1

2.9

excellent

1

4.9

excellent

13.3

good

2

7.1

excellent

3

9.6

excellent

4

12.9

good

= 3 repetitions

5

13.6

good

All others = 2 repetitions

Unfortunately, due to budget and time constrictions, which already affected our capacity to sequence the entire gene for all elephants in the study as it is mentioned in the text, we could not proceed to a more detailed investigation. Due to the low number of repetitions preformed and the lack of further validity testing as you suggest below, we agree it is important and necessary to perform further studies. Hopefully this initial study will bring to further use of this Point-of-care analyser and better knowledge on its use in Asian elephants.

Furthermore, similar studies were performed in guinea-pigs, in a specific animal strain that suffers of haemorrhagic fever viral diseases (by induction), leading to cytokine activation and vascular leakage, evolving therefore to activation of the coagulation cascade and terminating in DIC [1,2], which is what was also recently reported for the fatal cases of EEHV [3,4]. The VSPro was used in that study, and values obtained form now a new baseline for future comparisons on the study of this disease for this species and strain.

In an attempt also to use this analyser in Wallabies (n=14), the authors realised that all individuals presented a PT >35s, which is higher than the capacity of the device to read PT times.

Also: “The VSpro fibrinogen assay used here measures fibrinogen by thrombin-mediated enzymatic conversion to fibrin; thus, nothing precludes the use of this platform for other species. When our studies began, the assay was only validated for use in horses. However, studies are ongoing and the manufacturer has confirmed that the assay is now also validated for dogs.”[1]

Therefore, the VSPro analyser has proven to work for other species than cat, dog and horses, and was shown to not work in at least one species (the wallaby) for the measurement of aPTT. As our results fall into the reference range given by the manufacturer and inside the reading range of the machine, we believe that Asian elephant blood coagulation can also safely be analysed by this POC.

Point 3: It would also be recommended to dilute out some of the samples and see if you can receive a linear result and also determine if your PT and aPTT could give a linear response when mixed with calcium, kaolin, or another activator.

Response 3: Thank you for your suggestion. Those tests were not performed during this investigation. Nevertheless, we understand their importance and agree that they should be performed in the future.

A measurement of only one animal was performed filling a lower blood volume in the citrate tube (at 1/3 of normal volume), and therefore simulate the dilution effect. This individual presented a very prolonged coagulation time; PT>35s and aPTT>200s, meaning it overcame the maximum capacity of reading of the machine. Of course, no statistical conclusions can be made from this result.

For two individuals, we have over-filled the blood samples (with extra 1/3 than the recommended volume). In both animals it led to a faster coagulation, when compared to a normal filled tube of the same animals. Again, this was only tested in two animals, and therefore no statistical conclusions can be derived from this finding, but it shows at least the sensitivity of the machine to detect variances in the blood as it had less anticoagulant.

The following paragraph was included in the Conclusion section in Lines 603-608:

“Due to lack of financial, time, logistical and human resources it was not possible to run all the validation tests during this investigation. However, as a future perspective, we believe it would be an important topic for further research. Nevertheless, the large sample size used in this study and the results obtained are good indicators that this POC can be used in Asian elephants. These preliminary results are important for future clinical practice comparisons.”

Point 4: Some sort of validation is needed with this study. Perhaps even running PT/PTT/fibrinogen with other methods concurrently and seeing if the results align. Far too much stock is put into the results of this analyzer with no validation.

Response 4: Thank you for your comment. Yes, we agree that it would be the ideal.

Five elephants (all from the European population) had the coagulation parameters also measured by another laboratory. We run a Wilcoxon test comparing our VSPro results with the ones obtained from this laboratory and no significant differences were found between the two different methods. Nevertheless, the technique used by the external lab was also not validated for elephants, so a true comparison cannot be achieved. This is a common problem in non-model species.

However, this is what normally happens in a global scale; samples are sent to be tested in different laboratories, not validated, and using different techniques between countries/labs, restricting therefore possible comparisons. Still, and although a comparison with a validated external laboratory has not (yet) been possible, we believe that our large sample size, englobing several age classes, will give a better base of reference than non-validated techniques based on comparison many times with human ranges.

Point 5: Also, you cannot accept results when the analyzer was run at temperatures outside of Abaxis' guidelines, especially if no validation was performed to show that the results should not have been affected.

Response 5: The results found for the animals that were tested below or above the “ideal” temperatures provided by Abaxis fall into the average range of the others tested in room temperature. There was only one instance where the temperature forbidden the analysis as the machine “shut off” due to alert of high heat, and therefore no analysis was performed in the field, but later in the laboratory. For the eight Thai elephant samples from this camp, the temperature was limiting high, and their coagulation times could not be measured in the field and thus were analysed in the laboratory at room temperature. Likewise, the results obtained also fell within the range of values found for the tested population.

Only three individuals were analysed under 15°C. The animals in question were still at room temperature, as the enclosures where they spend the night are heated. The place of the cartridge analyse was colder, around 10°C. We do not know what their coagulation times would have been if they had been measured at room temperature, but once again, result values fall into the average range found for others tested at room temperature.

Temperature wise, the only noticeable difference was What was a slightly longer time for the machine to “warm up” and allow for cartridge introduction in colder weather, and faster readiness of the cartridge to accept the sample in warmer weather.

We believe it is important to highlight that we are working in field conditions with (endangered) wildlife species, which live in different countries and regions, with different temperatures, settings, and human handlings, and therefore, flexibility was needed, which is different from the situation in  a standardized laboratory working on small animal research.

Point 6: Perhaps you can run a single elephant’s sample in a warm room and at room temperature and a cold room at one of your research areas?

Response 6: Thank you for your great suggestion. Unfortunately, the study is over, and the samples have all been used up, so no more testing will be possible. When we devised and started the study, we were aware of the fact that the temperatures in the different settings (camps and zoos) may vary, but we were not aware of the large range these temperature differences may cover. For future studies, we will definitely consider this. Please see also our answer above.

Point 7: Additionally, PCV is known to be highly influential on coagulation parameters, so please report this for all animals included in the study.

Response 7: Thank you for your comment. Haematocrit was measured in a total of 31 individuals: four European elephants and 27 Thai elephants. They all fell into the range of values for normal healthy Asian elephants (average found was 36%, range 28-44%). No disease was reported in any of the sampled animals, so we assume that all individuals present a normal PCV and in the range suggested by the manufacture for better use of the equipment (15% < PCV value <65%).

Additional notes:

Point 8: The introduction discussing factor VII deficiencies doesn't specify that it is discussing it in humans at first and it is very confusing.

Response 8: To avoid future confusion, it has been changed in the text. Please see line 110.

Point 9: The introduction is quite clunky. I recommend reworking it and presenting it as you are trying to classify as much as you can about the coagulation cascade in Asian elephants.

Response 9: There is no evidence that the Asian elephants have any differences in their coagulation cascade compared with what is known from other mammals.

Nevertheless, the introduction has been edited. We hope it meets your expectations.

Point 10: Please justify comparing the Asian elephant factor VII gene sequence to that of African elephants. It is very likely the correct comparison, but it should be justified

Response 10: Thank you for your comment. It was chosen as it is the only F7 full gene information available to compare and as it was the one used by the study previously reporting coagulation disorder in Asian elephants due to mutation in this gene.  

Information has been added in the text, in the Method sections, Lines 256 -257:  

“Single nucleotide polymorphisms (SNPs) were evaluated for possible impact on the factor VII protein structure in comparison with complete F7 gene, which currently is only available for Loxodonta africana, as done by the reference study [5], and using the web-based protein variation effect predicting software packages SIFT [6] and PROVEAN v1.1.3 [7].”

Point 11: All of your insistence that coagulation testing should be done routinely with blood work and receiving more immediate coagulation test results when treating EEHV are stated without any support. How would having this information on a routine basis be helpful?

Response 11: Thank you for this comment. It is mentioned in the text (line 532 – foetal mummification) that for example fibrinogen can alter with ephemerons states, and aPTT can change during pregnancy for example (line 497-498). Other diseases can alter coagulation times and fibrinogen concentrations, and illness could be maybe early detected if this tool was used as part of a regular diagnostic routine.

The following paragraph was added to the introduction section, lines 69 – 87. We hope it meets your expectations:

“In addition to a complete blood count, coagulation time results and fibrinogen concentrations can be used as valuable and easily accessible health indicators, because stress, illness, injury, medications and surgery affect coagulation parameters [8]. Coagulation times provide information in a large variety of clinical ephemerons alterations, such as sepsis, hepatic disfunction, decrease of vitamin K, shock, trauma, embolism, platelet bleeding disorders, coagulation factory deficiency and disseminated intravascular coagulation (DIC) [8–11]. Liver disfunction may affect the coagulation cascade in several ways since this organ produces most of the coagulation factors and affects vitamin K absorption. Therefore, any illness affecting the liver, like inflammation, neoplasia, biliary statis and the use of chronic medication may lead to coagulation deficiency. Infectious diseases, severe systemic diseases or immune-mediated disease can also alter normal coagulation times. Due to this panoply of factors that may affect coagulation, it is suggested that coagulation times should be accessed as a pre-surgical test for any animal, regardless of age [8].

Fibrinogen is used as a specific and sensitive marker for inflammation for example in humans [12] and in horses, and its early recognition has shown to be essential for the diagnosis of diseases and proper treatment planning. In horses, fibrinogen serial testing provides information regarding treatment efficacy in length, prognosis in several infectious or inflammatory conditions such as pleuropneumonia, abdominal abscess, endometritis and endocarditis [8].

Point 12: How would having these results sooner help--would clinicians start plasma treatment sooner?

Response 12: This is an important comment, and the answer is that we still don’t know. There is no published information on the surveillance of coagulation alteration during an EEHV-HD episode using serial sampling. It would be of great value to follow the changes alongside the viremia and like a normal control monitoring, it might give us an idea of vascular disbalance even before the high titers are obvious or above the 5000VGE/mL threshold. We believe this should be further investigated.

At the moment, the continue monitoring of viral titers allows us for a maximum 1-2 weeks of recognition of increasing viremia and preparation for treatment. If somehow, for example, we realize that 3 weeks before an EEHV-HD event the coagulation times are abnormal, yes, plasma transfusion should be started to support coagulation by providing coagulation factors earlier.

Our study aims to publish, for the first time, coagulation data from elephants analysed with a fast-achieving method, which is also practical for the field. We hope these results serve as reference values for further (and much needed) studies in this area, especially those differentiating healthy animals from EEHV-HD diseased ones.

Point 13: Would it help with prognosis?

Response 13: Probably yes. No one knows at the moment as there are no continuous studies on coagulation in elephants suffering from haemorrhagic disease or any other disease (e.g. liver disease would be of interest and even lameness due to foot problem which is common in this species as well), but as it does in other species, it is expected to help prognosis.

If coagulation times alter in a time-dependent manner for all surviving / dying calves, it would be great information to assist the prognosis.

In guinea-pigs, PT and aPTT were increased when a viral haemorrhagic fever infection progressed past day 7 (measured using VSPro [2]). It would move our knowledge about the disease and treatment options tremendously forward if one were to run the same investigation for EEHV-HD in elephants.

Point 14: It seems concerning that calves recovered from EEHV would still have significantly different aPTT values. Do you actually believe this is the case, or do you believe this is spurious. Do you think there are long lasting coagulation effects of this disease process?

Response 14: Thank you for your question. We have also thought about this quite a lot. It is the most probable. As EEHV-HD is greatly misbalancing such an important system as the coagulation cascade, it will surely take quite a while to “reset” the vascular and coagulation systems.

With EEHV-HD, most of the pathological knowledge is novelty and based on necropsies, that showed that DIC and a gross vascular alteration are present. There is destruction of the microvasculature that leads to DIC, visually assessed for example with the cyanotic tongue and oedema of the face and limbs [3]. Long lasting coagulation effects of the disease in the survival cases are still to be studied. But we expect the timeframe from a DIC scenario to a complete recovered status, to cover at least several weeks or even months.

Nevertheless, results obtained in this study for survival cases are based on a low sample size, therefore, even though a prolonged recovery period seems very plausible, this too need to be further investigated in the future.

Point 15: I understand how in field situations performing an estimated platelet count on a slide is what needs to occur, but I would specify that you are performing an estimated platelet count and I would delve more into the methods that hopefully made this a very standardized process--did you invert the tube a certain amount of times prior to withdrawing the blood for the slide? Did you check the feathered edge for platelet clumps and determine that the slides were free of them?

Response 15: Thank you for your comment. Yes, all blood smears were performed in a standardized process and the feathered edge of the blood smear was analysed. Small clots were visible on a few occasions, nevertheless it is a common finding for elephants. When no major clotting was noticed, the blood smears were accepted in the study to estimate the platelet count.

Cited References:

  1. Condrey, J.A.; Flietstra, T.; Nestor, K.M.; Schlosser, E.L.; Coleman-Mccray, J.D.; Genzer, S.C.; Welch, S.R.; Spengler, J.R. Prothrombin time, activated partial thromboplastin time, and fibrinogen reference intervals for inbred strain 13/n guinea pigs (Cavia porcellus) and validation of low volume sample analysis. Microorganisms 2020, 8, 1–11, doi:10.3390/microorganisms8081127.
  2. Mendenhall, M.; Russell, A.; Smee, D.F.; Hall, J.O.; Skirpstunas, R.; Furuta, Y.; Gowen, B.B. Effective oral favipiravir (T-705) therapy initiated after the onset of clinical disease in a model of arenavirus hemorrhagic fever. PLoS Negl. Trop. Dis. 2011, 5, 1–10, doi:10.1371/journal.pntd.0001342.
  3. Guntawang, T.; Sittisak, T.; Kochagul, V.; Srivorakul, S.; Photichai, K.; Boonsri, K.; Janyamethakul, T.; Boonprasert, K.; Langkaphin, W.; Thitaram, C.; et al. Pathogenesis of hemorrhagic disease caused by elephant endotheliotropic herpesvirus (EEHV) in Asian elephants (Elephas maximus). Sci. Rep. 2021, 11, 1–13, doi:10.1038/s41598-021-92393-8.
  4. Perrin, K.; Kristensen, A.; Bertelsen, M.; Denk, D. Retrospective review of 27 European cases of fatal elephant endotheliotropic herpesvirus ‑ haemorrhagic disease reveals evidence of disseminated intravascular coagulation. Sci. Rep. 2021, 1–13, doi:10.1038/s41598-021-93478-0.
  5. Lynch, M.; McGrath, K.; Raj, K.; McLaren, P.; Payne, K.; McCoy, R.; Giger, U. Hereditary factor VII deficiency in the Asian elephant (Elephas maximus) caused by a F7 missense mutation. J. Wildl. Dis. 2017, 53, 248–257, doi:c.
  6. Vaser, R.; Adusumalli, S.; Leng, S.N.; Sikic, M.; Ng, P.C. SIFT missense predictions for genomes. Nat. Protoc. 2016, 11, 1–9, doi:10.1038/nprot.2015.123.
  7. Choi, Y.; Chan, A.P. PROVEAN web server: A tool to predict the functional effect of amino acid substitutions and indels. Bioinformatics 2015, 31, 2745–2747, doi:10.1093/bioinformatics/btv195.
  8. Zoetis VetScan Pro - utilization guide Available online: https://www.zoetisus.com/products/diagnostics/vetscan/pdf/vetscan-vspro-utilization-guide.pdf (accessed on Oct 11, 2021).
  9. Palta, S.; Saroa, R.; Palta, A. Overview of the coagulation system. Indian J. Anaesth. 2014, 58, 515–523, doi:10.4103/0019-5049.144643.
  10. Zehnder, J.L.; Leung, L.L.; Landaw, S.A. Clinical use of coagulation tests Available online: https://somepomed.org/articulos/contents/mobipreview.htm?14/27/14769.
  11. Fasano, A.; Sequeira, A. Blood coagulation; 2017; Vol. 18; ISBN 9783319605135.
  12. Davalos, D.; Akassoglou, K. Fibrinogen as a key regulator of inflammation in disease. Semin. Immunopathol. 2012, 34, 43–62, doi:10.1007/s00281-011-0290-8.

Reviewer 2 Report

This manuscript describes a tool to generate coagulation data from elephants quickly, as well as a comparison of coagulation parameters in Asian elephants in Asia and Europe. Additionally, it documents sequence alterations in the F7 gene, in which a mutation was previously identified in an Asian elephant. I recommend publication with some edits, because this new information is beneficial for future studies. 

The manuscript will benefit from a close read and edit.  For instance, lines 102 and 103 lists dog breeds, but not a complete sentence.

In terms of content, a few additions will improve the manuscript. One thing that is unclear to me is how this information about coagulation can be used during emergencies or with diseased animals. The authors state that it will be useful, but they do not list any examples. An explanation of how these specific factors that they measured in this study change in disease situations will be helpful. If they don't change, then mentioning that will also be helpful, as long as it states when and why these coagulation parameters should be measured.

The methods state that different types of blood collection were performed, like the use of different needles. It also states that whenever possible PT and aPTT were measured within 30 minutes. I'm wondering if any differences in the results were observed between the different needles used or for samples that were measured later than 30 min. Since I think the documentation of this instrument to measure elephant PT and aPTT is important, it is also important to document any differences observed related to sample collection and timing of measurement, which could benefit future studies.

Last, while the sequencing of F7 is important to share with the research community, it is also important to point out that other genes are documented to cause deficiencies in coagulation. Sequencing only F7, and not other known coagulation deficiency genes, represents a limitation of this study.

Author Response

Dear reviewers,

We are very thankful for your review and comments on our manuscript. We have conducted extensive revisions in agreement with your inputs and we believe the manuscript has been greatly improved. The line numbers referenced below correspond to the “simple markup” option, inside the tab “Review”> Track changes. You can follow all details of the changes made to the manuscript if you select the “all markup” option in the same tab. We thank you greatly for your time, corrections, and suggestions.

Please find the replies to your comments below.

Reviewer 2

This manuscript describes a tool to generate coagulation data from elephants quickly, as well as a comparison of coagulation parameters in Asian elephants in Asia and Europe. Additionally, it documents sequence alterations in the F7 gene, in which a mutation was previously identified in an Asian elephant. I recommend publication with some edits because this new information is beneficial for future studies. 

Point 1: The manuscript will benefit from a close read and edit.  For instance, lines 102 and 103 lists dog breeds, but not a complete sentence.

Response 1: Thank you for your comment and noticing, it has been changed in the text, now at line 124-125.

Point 2: In terms of content, a few additions will improve the manuscript. One thing that is unclear to me is how this information about coagulation can be used during emergencies or with diseased animals. The authors state that it will be useful, but they do not list any examples. An explanation of how these specific factors that they measured in this study change in disease situations will be helpful.  If they don't change, then mentioning that will also be helpful, as long as it states when and why these coagulation parameters should be measured.
Response 2: Thank you for your comment and great suggestion. The below paragraph was added to the introduction section, lines 69 – 87. We hope it meets your expectations.

“In addition to a complete blood count, coagulation time results and fibrinogen concentrations can be used as valuable and easily accessible health indicators, because stress, illness, injury, medications and surgery affect coagulation parameters [2]. Coagulation times provide information in a large variety of clinical ephemerons alterations, such as sepsis, hepatic disfunction, decrease of vitamin K, shock, trauma, embolism, platelet bleeding disorders, coagulation factory deficiency and disseminated intravascular coagulation (DIC) [2–5]. Liver disfunction may affect the coagulation cascade in several ways since this organ produces most of the coagulation factors and affects vitamin K absorption. Therefore, any illness affecting the liver, like inflammation, neoplasia, biliary statis and the use of chronic medication may lead to coagulation deficiency. Infectious diseases, severe systemic diseases or immune-mediated disease can also alter normal coagulation times. Due to this panoply of factors that may affect coagulation, it is suggested that coagulation times should be accessed as a pre-surgical test for any animal, regardless of age [2]. Fibrinogen is used as a specific and sensitive marker for inflammation for example in humans [6] and in horses, and its early recognition has shown to be essential for the diagnosis of diseases and proper treatment planning. In horses, fibrinogen serial testing provides information regarding treatment efficacy in length, prognosis in several infectious or inflammatory conditions such as pleuropneumonia, abdominal abscess, endometritis and endocarditis [2].”

Point 3: The methods state that different types of blood collection were performed, like the use of different needles. It also states that whenever possible PT and aPTT were measured within 30 minutes. I'm wondering if any differences in the results were observed between the different needles used or for samples that were measured later than 30 min. Since I think the documentation of this instrument to measure elephant PT and aPTT is important, it is also important to document any differences observed related to sample collection and timing of measurement, which could benefit future studies.

Response 3: Thank you for your important comment. For only eight Thai elephants the temperature was limiting high, and their coagulation times could not be measured in the field and thus were analysed in the laboratory at room temperature. The results obtained fell within the range of values found for the tested population.

No coagulation time differences were observed for the different collection methods, and for all blood samples, needle gauge used was always higher than 23 G, which is the threshold to avoid coagulation activation during collection. For instance, even though for cats, dogs, and humans different methods of blood collection are applied, coagulation tests are still performed with high reliability and reproducibility. In our elephant study, only two Zoos generally used butterfly catheters, in a few others butterfly catheters were used only for calves, but the gross majority collected blood using a pink or larger needle copulated to a syringe.

Point 4: Last, while the sequencing of F7 is important to share with the research community, it is also important to point out that other genes are documented to cause deficiencies in coagulation. Sequencing only F7, and not other known coagulation deficiency genes, represents a limitation of this study.

Response 4: Thank you for your comment, it is very relevant.

Unfortunately, for elephants only the F7 is known so far to cause coagulation disorder in elephants. You are correct, other coagulation factor genes should be investigated, but deep research on a molecular (gene sequence) level is usually only performed once a coagulation disorder has been detected, which is normally only the case when coagulation times are measured. Our aim here, regarding the F7 gene sequence investigation was to confirm/investigate if individuals in the Thai and European populations would be carriers of a previously reported hereditary disease found in Asian elephants, and which comprised a mutation in the coagulation factor VII.

The following paragraph has been added to the introduction section (Line 153-155).

“Although other coagulation factor deficiencies might be present, this investigation focuses only on the detection of a previously reported hereditary disorder involving factor VII in Asian elephants.”
